# OpenReview forum: "Combining Cost Constrained Runtime Monitors for AI Safety"
_NeurIPS.cc/2025/Conference — NeurIPS 2025 poster_

### Official Review · Reviewer_RTmL · 2025-06-10

**Clarity:** 3
**Significance:** 2
**Originality:** 3
**Rating:** 4
**Confidence:** 3

**Summary:**

This paper addresses the challenge of optimising safety interventions in AI systems under cost constraints by combining runtime monitors. The authors formulate the problem as a partially observable Markov decision process (POMDP), where the goal is to learn a policy that decides when and which monitors to invoke, and when to audit outputs based on monitor scores, in order to maximise recall on misaligned outputs while respecting a fixed budget.

They evaluate their framework on BigCodeBench, focusing on detecting backdoored code samples. Results show that a stochastic monitoring policy, which may audit without always querying a monitor, achieves higher recall under the same cost compared to deterministic baselines that always call monitors. Furthermore, they demonstrate that a stochastic combination of two monitors can strictly outperform all single-monitor policies, including those that always invoke both monitors.

**Questions:**

Given the limitations of your approach (discussed under Cons above), how common are setups in which your approach is applicable and useful?

**Ethical Concerns:**

["NO or VERY MINOR ethics concerns only"]

**Final Justification:**

During the discussion period, the authors made a case for the practical applicability of their work despite scalability issues. I appreciate their perspective and therefore maintain my positive assessment. However, the cons in my initial review remain. In particular, regarding the authors' OOD experiment, it is unclear how representative it is of real-world misalignment. The drop in safety score from 82% to 48% is substantial, raising concerns about the method’s absolute effectiveness under realistic distribution shifts. Also. although I appreciate the authors' explanation of the practical utility of two monitors, the technique's scalability limitations persist.

**Limitations:**

yes

**Paper Formatting Concerns:**

no concerns

**Quality:**

2

**Strengths And Weaknesses:**

Pros:
=====
* The paper introduces a mathematical formulation (as a POMDP) for combining runtime monitors under budget constraints.
* It also shows that smart combinations of monitors can achieve higher recall at the same or lower cost.
* The evaluation is done on a realistic setting: backdoor detection in AI-generated code (BigCodeBench).
* The paper is generally well-written. There are small typos in places, e.g. use of \PI_{TM} in Figure 5, but \PI_{BM} in the text describing it.

Cons:
=====
* The paper's method relies on the distribution of misaligned examples being stable. This comes from the training data and is assumed to reflect the real-world distribution the system will face during deployment. However, in practice distributions shift, which may negatively impact the audit decisions.
* Finding the optimal monitoring protocol requires enumerating all possible monitor calling strategies. This brute force search is intractable beyond 4 monitors, and the paper doesn't explore any approximate or scalable learning method.

Both my concerns are acknowledged by the authors under limitations. However, I'm not sure if this makes it better. They are still pretty serious limitations that make me doubt the practical applicability of the proposed technique.

---

> ### Author Rebuttal · Authors · 2025-07-30
>
> We thank reviewers for their comments. We’ll first give a general overview of the paper’s strengths and answer some common objections. Then dive into answering specific questions.
>
> ## Significance of our work
>
> Our paper is focused on optimally combining information from runtime monitors with different cost and performance profiles. We study how to rigorously construct these protocols agnostic to the specific setting. This is an increasingly important problem in AI safety. Since the Neurips submission deadline, OpenAI has announced that they use a combination of two monitors for ChatGPT Agent’s bio-misuse mitigation, and Anthropic researchers put out a new paper on using multiple probe-based monitors with varied costs. However, neither paper presents a principled way of combining monitors:
>
> Cunningham et al.’s “Cost-Effective Constitutional Classifiers via Representation Re-use” tests various probe-based monitors with different overhead computational costs. They combine monitors using the simple heuristic of “use the cheap monitor first, and if suspicious, use the expensive monitor. ” Instead of finding the optimal region to call the second monitor at a set of budgets to sketch out the cost curve, they simply sweep over the threshold on when to call the second monitor to vary spend (Figure 4). Instead of using likelihood ratios to combine the two monitors when two are called, they only use data from the second monitor.
>
> OpenAI’s “ChatGPT Agent System Card” section 5.2.2.2. also mentions the use of a monitoring system consisting of two monitors. Their first monitor checks if the content is related to biology. If so, a second monitor checks whether the response or prompt contains hazardous information. This shows that monitoring protocols constructed from multiple monitors are already used in production AI systems for AI safety.
>
> To be clear, it’s possible that our optimized algorithm will also recommend a final strategy of “use cheap monitor first, then use expensive monitor, and only rely on expensive monitor’s scores if they’re available.” It’s also possible that practitioners have some strong prior over which monitors are more robust and trustworthy, and choose a simpler or regularized strategy in lieu of something fitted to the data like the one we proposed. However, we believe it’s important to showcase how to construct these protocols in a rigorous, data-driven manner.
>
> The existing literature also relies on heuristics to construct these protocols. McKenzie et al.’s “Detecting High-Stakes Interactions with Activation Probes” chooses when to call the second monitor based on how close the first monitor’s score is to the median (as a heuristic measure for uncertainty). Parrack et al.’s “Benchmarking Deception Probes via Black-to-White Performance Boosts” and McKenzie et al. both take a simple weighted average of multiple monitors to make the final auditing decision, as opposed to using likelihood ratio methods. We hope reviewers will reconsider the timeliness and importance of our work given these new papers, and we will incorporate these citations into our paper.
>
> ## New experiment addressing OOD shift and diverse settings
>
> Reviewers also brought up the lack of diverse settings in our experiments, how our “malicious” code from o1-mini is not sufficiently obscured, and how distributional shifts could reduce the performance of the protocol. We do think distributional shift could be a serious issue: if we overfit to training data, we might do worse on real-world deployments. However, the right comparison is “Are we doing better than not fitting at all?” We ran an additional experiment where we took the thresholds fit to the o1-mini generated backdoors on BigCodeBench, and tested the monitoring protocol’s performance on a random sample of backdoored code from Wen et al. “Adaptive Deployment of Untrusted LLMs Reduces Distributed Threats.” These are backdoored code generated using a much more optimized pipeline on the APPS dataset, which acts as a good out-of-distribution test.
>
> While the overall safety scores decrease significantly, we find that the order in which policies do the best remains the same across all experiments. For the main result in Figure six, when the total budget is 1.5, $\Pi_\text{SR}$’s safety decreased from 82% to 48% on the OOD set, but it still outperforms always using both monitors (24%), or using only the 70B model (32% safety) or 8B model (39% safety). The results are also similar in single monitor settings, where $\Pi_\text{NPS}$ outperforms $\Pi_\text{NP}$, which outperforms $\Pi_\text{Naive}$. Thus, for this specific distribution shift, it was better that we had fitted our thresholds to the data. Future work could test distribution shifts that are more likely to occur for specific settings that we’re constructing monitoring protocols for. At runtime, practitioners could also continue to incorporate new data into the training set and tune the thresholds to keep them updated.
>
> Generally, we think the strength of our paper lies in our principled and rigorous theoretical framework. The important comparison is “Does our paper provide a better way of combining monitors compared to heuristic baselines?” Given that most policy classes search over options which include the heuristic baselines (such as “call the cheap monitor first and the expensive monitor if it’s above a certain threshold”), we think the answer is trivially yes.
>
> ## Adding a new guide on how to apply our methods
>
> Finally, reviewer f693 and RTmL have found our work cleanly written (thank you!). Our goal is to provide an easy-to-follow guide to real-world practitioners building monitoring protocols. Given the rise in systems combining probe-based monitors and black box monitors, we added a guide that walks through an example on how to combine probe monitors and black box monitors using data from Parrack et al., making our work even more approachable.
>
> ## Additional responses for reviewer RTmL
>
> **Re distribution shift**: See above.
>
> **Re scalability issues & how common our setup is**: We do not explore cases with more than four monitors because we think they are well served by existing literature on costly classification trees or costly classification RL (see lines 70-74 and 293-296). Again, existing monitoring systems are mostly using two monitors and clumsily combining them, and thus we think this paper addresses an important and timely problem. Systems using two monitors are already used for ChatGPT Agent, and we think they’ll only become more commonplace in the future.

---

> > ### Comment · Reviewer_RTmL · 2025-08-05
> >
> > Thank you for the detailed response. It does cover my queries.

---

> > > ### Comment · Area_Chair_qYaZ · 2025-08-06
> > >
> > > Dear reviewer, how does the answer to your questions impact your overall view of the paper, including your ratings?

---

### Official Review · Reviewer_3wd1 · 2025-07-01

**Clarity:** 2
**Significance:** 1
**Originality:** 2
**Rating:** 4
**Confidence:** 4

**Summary:**

This paper considers the budgeting problem of when to call a more expensive monitor when trying to detect AI misalignment. This paper proposes approach for determining this threshold based on some training data and a fixed budget. The proposed approach is empirically evaluated on a single task of detecting whether or not an AI system has output code with a backdoor or not. The authors show that there approach provides a Pareto improvement over just using a single monitor.

**Questions:**

* Why is there just a fixed budget and how would we know what this is in practice?
* Adding Gaussian noise to the monitor scores seems like a terrible way to make them continuous. The more reasonable way to do this is to extract the logits for each of the digits from the output layer of the Llama model (either 8B or 70B) and compute a weighted expectation from that categorical distribution (over the tokens "0", "1", ..., "9") by applying softmax to the logits.
* Why is this an important problem that warrants an entire paper about it?
* Why are the two thresholds referred to as a policy? What are the states and actions?
* Are there other problems where this is more important? Why choose backdoors in code as the simulation of deception? Is this based on the prior AI control work only?
* Is the plan for this monitoring system to be deployed at test-time? If so, it seems like a fixed budget makes no sense. This is a long running system with many outputs. Or is the budget more of a suggestion or intended for a specific time window

**Ethical Concerns:**

["NO or VERY MINOR ethics concerns only"]

**Final Justification:**

The authors have more than adequately addressed my concerns.

**Limitations:**

Yes

**Quality:**

2

**Strengths And Weaknesses:**

### Strengths
* The paper presents a simple and easy-to-understand budgeting strategy for choosing when to call an additional more expensive monitor.
* The result of the dip in log-likelihood where higher monitor scores do not necessarily correspond to higher precision (Figure 2a) is very interesting.
* The results on this limited setup are fairly good and shows the benefit of this approach in a limited toy setting.

### Weaknesses
* The paper is not well motivated in the text. The paper somewhat assumes that the reader is familiar with prior AI control and monitoring work, and it just assumes that the importance of this problem is self-evident. Monitoring is obviously important for AI safety, but why is this problem important in particular? The text does not answer this question adequately. (I believe this problem is important - this is more about the writing.)
* The experiments are quite limited. The approach is evaluated in one scenario on one dataset, with only two external monitors. Section 4 just shows that the large expensive monitor works fairly well at detecting backdoors in code (not even that great though). Section 6 shows how to trade-off two monitors.
* A massive limitation acknowledged by the authors is that the learned threshold might not generalize to out-of-distribution data, but they make no effort to try this on more than one scenario.
* The toy scenario is somewhat fabricated especially since o1 is not being told to obfuscate the backdoors it is creating. This is a very weak simulation of deception.
* The proposed approach is made to seem much more profound and complicated than it is in actuality. For example, the authors refer to a learned threshold for choosing whether or not to call an additional more powerful monitor, and depending on the results, which outputs should be audited (essentially the most powerful monitor) a policy. They also describe this general enumeration of all possible monitor combinations, which they do not even do themselves in their experiments. Calling the choice of which monitors to call a POMDP is quite the stretch.
* Furthermore, the final approach is not so sophisticated and seems like the first thing someone would try when given this problem: just learn a threshold for when the next most expensive monitor (auditing is just a very expensive and very accurate monitor) should be called based on some training data.

---

> ### Author Rebuttal · Authors · 2025-07-30
>
> We thank reviewers for their comments. We’ll first give a general overview of the paper’s strengths and answer some common objections. Then dive into answering specific questions.
>
> ## Significance of our work
>
> Our paper is focused on optimally combining information from runtime monitors with different cost and performance profiles. We study how to rigorously construct these protocols agnostic to the specific setting. This is an increasingly important problem in AI safety. Since the Neurips submission deadline, OpenAI has announced that they use a combination of two monitors for ChatGPT Agent’s bio-misuse mitigation, and Anthropic researchers put out a new paper on using multiple probe-based monitors with varied costs. However, neither paper presents a principled way of combining monitors:
>
> Cunningham et al.’s “Cost-Effective Constitutional Classifiers via Representation Re-use” tests various probe-based monitors with different overhead computational costs. They combine monitors using the simple heuristic of “use the cheap monitor first, and if suspicious, use the expensive monitor. ” Instead of finding the optimal region to call the second monitor at a set of budgets to sketch out the cost curve, they simply sweep over the threshold on when to call the second monitor to vary spend (Figure 4). Instead of using likelihood ratios to combine the two monitors when two are called, they only use data from the second monitor.
>
> OpenAI’s “ChatGPT Agent System Card” section 5.2.2.2. also mentions the use of a monitoring system consisting of two monitors. Their first monitor checks if the content is related to biology. If so, a second monitor checks whether the response or prompt contains hazardous information. This shows that monitoring protocols constructed from multiple monitors are already used in production AI systems for AI safety.
>
> To be clear, it’s possible that our optimized algorithm will also recommend a final strategy of “use cheap monitor first, then use expensive monitor, and only rely on expensive monitor’s scores if they’re available.” It’s also possible that practitioners have some strong prior over which monitors are more robust and trustworthy, and choose a simpler or regularized strategy in lieu of something fitted to the data like the one we proposed. However, we believe it’s important to showcase how to construct these protocols in a rigorous, data-driven manner.
>
> The existing literature also relies on heuristics to construct these protocols. McKenzie et al.’s “Detecting High-Stakes Interactions with Activation Probes” chooses when to call the second monitor based on how close the first monitor’s score is to the median (as a heuristic measure for uncertainty). Parrack et al.’s “Benchmarking Deception Probes via Black-to-White Performance Boosts” and McKenzie et al. both take a simple weighted average of multiple monitors to make the final auditing decision, as opposed to using likelihood ratio methods. We hope reviewers will reconsider the timeliness and importance of our work given these new papers, and we will incorporate these citations into our paper.
>
> ## New experiment addressing OOD shift and diverse settings
>
> Reviewers also brought up the lack of diverse settings in our experiments, how our “malicious” code from o1-mini is not sufficiently obscured, and how distributional shifts could reduce the performance of the protocol. We do think distributional shift could be a serious issue: if we overfit to training data, we might do worse on real-world deployments. However, the right comparison is “Are we doing better than not fitting at all?” We ran an additional experiment where we took the thresholds fit to the o1-mini generated backdoors on BigCodeBench, and tested the monitoring protocol’s performance on a random sample of backdoored code from Wen et al. “Adaptive Deployment of Untrusted LLMs Reduces Distributed Threats.” These are backdoored code generated using a much more optimized pipeline on the APPS dataset, which acts as a good out-of-distribution test.
>
> While the overall safety scores decrease significantly, we find that the order in which policies do the best remains the same across all experiments. For the main result in Figure six, when the total budget is 1.5, $\Pi_\text{SR}$’s safety decreased from 82% to 48% on the OOD set, but it still outperforms always using both monitors (24%), or using only the 70B model (32% safety) or 8B model (39% safety). The results are also similar in single monitor settings, where $\Pi_\text{NPS}$ outperforms $\Pi_\text{NP}$, which outperforms $\Pi_\text{Naive}$. Thus, for this specific distribution shift, it was better that we had fitted our thresholds to the data. Future work could test distribution shifts that are more likely to occur for specific settings that we’re constructing monitoring protocols for. At runtime, practitioners could also continue to incorporate new data into the training set and tune the thresholds to keep them updated.
>
> Generally, we think the strength of our paper lies in our principled and rigorous theoretical framework. The important comparison is “Does our paper provide a better way of combining monitors compared to heuristic baselines?” Given that most policy classes search over options which include the heuristic baselines (such as “call the cheap monitor first and the expensive monitor if it’s above a certain threshold”), we think the answer is trivially yes.
>
> ## Adding a new guide on how to apply our methods
>
> Finally, reviewer f693 and RTmL have found our work cleanly written (thank you!). Our goal is to provide an easy-to-follow guide to real-world practitioners building monitoring protocols. Given the rise in systems combining probe-based monitors and black box monitors, we added a guide that walks through an example on how to combine probe monitors and black box monitors using data from Parrack et al., making our work even more approachable.
>
> ## Additional responses for reviewer 3wd1
>
> **Re motivating the paper**: This is an important point! We’ve addressed this mostly above and emphasize that current systems using multiple monitors are not combining them effectively.
>
> **Re more experiments and out of distribution robustness**: see additional experiments above. The main criteria to validate our method in terms of out of distribution generalization is if we think we’ll get worse performance by fitting the model to the data. If we are worried, we could always regularize our fit more by increasing kernel width or enforcing monotonicity constraints, and we will change the paper to include some ideas here.
>
> We think the experiments are mostly illustrative, since the gains to optimizing the monitor weighting is extremely monitor-dependent, a fact that doesn’t change even if we tried more monitors and scenarios. The main goal of the experiments is to illustrate and communicate how to apply these methods.
>
> **Re “enumeration of all possible monitor combinations, which they do not even do themselves in their experiments”**: We believe that we have enumerated all possible monitor combinations (subject to calling the additional monitor in one contiguous region) in the main experiment for Figure six? We swept through 200 distinct thresholds (see algorithm 1 in appendix section A.5, and also the FullNPOneRegion class in strategy_class/new_strategies.py in supplemental materials). If we open ourselves to calling the monitor in non-contiguous regions (e.g., two intervals), we doubt that would meaningfully add additional safety margins.
>
> **Re sophistication and RL-language:** We agree that this overall approach is not very sophisticated, and we’re just learning a fairly simple decision rule. We thought that terms from RL were more appropriate, since it’s natural to condition our final safety and cost metrics on some “policy,” especially since the number of parameters for that policy can change depending on the number of monitors. The hidden state in this POMDP is the alignment of the output, and the actions are (1) call an uncalled monitor, (2) audit or (3) terminate without auditing (this is also discussed in the Monitoring as POMDP section on line 108 through 115). As mentioned above, even this “simple” method for choosing how to combine two monitors is not currently adopted by the literature.
>
> **Re fixed budget**: our budget is the average cost per output from the untrusted model (see line 103). For every output at test time, we’ll assume that the AI company is willing to spend some amount of cost on monitoring. We solve the optimization problem for a range of budgets, so companies can see what the returns are to spending more on monitoring. Practitioners can also solve the dual cost minimization problem (i.e., how much they need to spend to achieve a desired level of safety).
>
> **Re gaussian noise & Control setting**: We agree that softmax over logits would be better, but we also think that the value of our comes mostly from the framework, and that this paper wouldn’t lose too much value even if we used entirely simulated monitors scores instead of real world scores. We chose the control setting mostly for idiosyncratic reasons. Any setting where you have multiple monitors could be used here. Indeed, our new added guide uses probe and black box monitors for standard deception tasks following Parrack et al’s paper.

---

> > ### Comment · Reviewer_3wd1 · 2025-08-05
> >
> > Thank you for your thorough and detailed response. I am happy to raise my score!

---

### Official Review · Reviewer_Mmt4 · 2025-07-03

**Clarity:** 2
**Significance:** 2
**Originality:** 2
**Rating:** 4
**Confidence:** 2

**Summary:**

This paper develops a formal framework and practical protocol for optimally combining multiple AI runtime monitors under a unified cost constraint, using likelihood-ratio-based decision policies and exhaustive search to maximize recall for detecting misaligned model outputs while respecting budget limits; the approach is validated on code backdoor detection using Llama 3.1 70B/8B monitors and demonstrates strong empirical improvements over baseline strategies.

**Questions:**

1. How would your approach scale to settings with more monitors or more complex invocation strategies (e.g., more than 4 monitors, hierarchical calls)?

2. Could you evaluate the performance of the proposed technique under strong adversarial distribution shifts or monitors with significant miscalibration?

3. Can you clarify the computational/latency overhead for the protocol in a real-time deployment scenario?

**Ethical Concerns:**

["NO or VERY MINOR ethics concerns only"]

**Final Justification:**

After considering the author’s rebuttal and new OOD experiments, I have decided to slightly raise my score. The response addressed most of the concerns about robustness to distribution shift, practical latency, and real-world relevance, showing that the proposed method can stably outperform baselines. However, scalability to many monitors remains limited, and further empirical analysis of failure cases and monitor miscalibration would strengthen the work.

**Limitations:**

Yes.

**Paper Formatting Concerns:**

None.

**Quality:**

2

**Strengths And Weaknesses:**

Strengths:

+ this paper introduces , general method for cost-sensitive composition of multiple runtime monitors using rigorous likelihood ratio theory.

+ this paper provides a thorough empirical analysis and ablations showing that combining monitors yields superior safety-recall trade-offs compared to single-monitor or naïve approaches.

Weakness:

+ The scalability of the proposed method is limited, as exhaustive policy search is only tractable for a small number of monitors and simple invocation strategies.

+ The method’s robustness under severe distribution shift or with more diverse/complex monitors is not fully evaluated.

+ Qualitative failure cases and breakdowns for different attack types are limited; more analysis would increase transparency.

---

> ### Author Rebuttal · Authors · 2025-07-30
>
> We thank reviewers for their comments. We’ll first give a general overview of the paper’s strengths and answer some common objections. Then dive into answering specific questions.
>
> ## Significance of our work
>
> Our paper is focused on optimally combining information from runtime monitors with different cost and performance profiles. We study how to rigorously construct these protocols agnostic to the specific setting. This is an increasingly important problem in AI safety. Since the Neurips submission deadline, OpenAI has announced that they use a combination of two monitors for ChatGPT Agent’s bio-misuse mitigation, and Anthropic researchers put out a new paper on using multiple probe-based monitors with varied costs. However, neither paper presents a principled way of combining monitors:
>
> Cunningham et al.’s “Cost-Effective Constitutional Classifiers via Representation Re-use” tests various probe-based monitors with different overhead computational costs. They combine monitors using the simple heuristic of “use the cheap monitor first, and if suspicious, use the expensive monitor. ” Instead of finding the optimal region to call the second monitor at a set of budgets to sketch out the cost curve, they simply sweep over the threshold on when to call the second monitor to vary spend (Figure 4). Instead of using likelihood ratios to combine the two monitors when two are called, they only use data from the second monitor.
>
> OpenAI’s “ChatGPT Agent System Card” section 5.2.2.2. also mentions the use of a monitoring system consisting of two monitors. Their first monitor checks if the content is related to biology. If so, a second monitor checks whether the response or prompt contains hazardous information. This shows that monitoring protocols constructed from multiple monitors are already used in production AI systems for AI safety.
>
> To be clear, it’s possible that our optimized algorithm will also recommend a final strategy of “use cheap monitor first, then use expensive monitor, and only rely on expensive monitor’s scores if they’re available.” It’s also possible that practitioners have some strong prior over which monitors are more robust and trustworthy, and choose a simpler or regularized strategy in lieu of something fitted to the data like the one we proposed. However, we believe it’s important to showcase how to construct these protocols in a rigorous, data-driven manner.
>
> The existing literature also relies on heuristics to construct these protocols. McKenzie et al.’s “Detecting High-Stakes Interactions with Activation Probes” chooses when to call the second monitor based on how close the first monitor’s score is to the median (as a heuristic measure for uncertainty). Parrack et al.’s “Benchmarking Deception Probes via Black-to-White Performance Boosts” and McKenzie et al. both take a simple weighted average of multiple monitors to make the final auditing decision, as opposed to using likelihood ratio methods. We hope reviewers will reconsider the timeliness and importance of our work given these new papers, and we will incorporate these citations into our paper.
>
> ## New experiment addressing OOD shift and diverse settings
>
> Reviewers also brought up the lack of diverse settings in our experiments, how our “malicious” code from o1-mini is not sufficiently obscured, and how distributional shifts could reduce the performance of the protocol. We do think distributional shift could be a serious issue: if we overfit to training data, we might do worse on real-world deployments. However, the right comparison is “Are we doing better than not fitting at all?” We ran an additional experiment where we took the thresholds fit to the o1-mini generated backdoors on BigCodeBench, and tested the monitoring protocol’s performance on a random sample of backdoored code from Wen et al. “Adaptive Deployment of Untrusted LLMs Reduces Distributed Threats.” These are backdoored code generated using a much more optimized pipeline on the APPS dataset, which acts as a good out-of-distribution test.
>
> While the overall safety scores decrease significantly, we find that the order in which policies do the best remains the same across all experiments. For the main result in Figure six, when the total budget is 1.5, $\Pi_\text{SR}$’s safety decreased from 82% to 48% on the OOD set, but it still outperforms always using both monitors (24%), or using only the 70B model (32% safety) or 8B model (39% safety). The results are also similar in single monitor settings, where $\Pi_\text{NPS}$ outperforms $\Pi_\text{NP}$, which outperforms $\Pi_\text{Naive}$. Thus, for this specific distribution shift, it was better that we had fitted our thresholds to the data. Future work could test distribution shifts that are more likely to occur for specific settings that we’re constructing monitoring protocols for. At runtime, practitioners could also continue to incorporate new data into the training set and tune the thresholds to keep them updated.
>
> Generally, we think the strength of our paper lies in our principled and rigorous theoretical framework. The important comparison is “Does our paper provide a better way of combining monitors compared to heuristic baselines?” Given that most policy classes search over options which include the heuristic baselines (such as “call the cheap monitor first and the expensive monitor if it’s above a certain threshold”), we think the answer is trivially yes.
>
> ## Adding a new guide on how to apply our methods
>
> Finally, reviewer f693 and RTmL have found our work cleanly written (thank you!). Our goal is to provide an easy-to-follow guide to real-world practitioners building monitoring protocols. Given the rise in systems combining probe-based monitors and black box monitors, we added a guide that walks through an example on how to combine probe monitors and black box monitors using data from Parrack et al., making our work even more approachable.
>
> ## Additional responses for reviewer Mmt4
>
> **Re limited scalability in the number of monitors**: We think the strength of our method is that it achieves optimality (assuming that you don’t call your second monitor in non contiguous regions), and we think the case with few monitors is actually much more likely to be used in real life as seen in the new OpenAI and Cunningham et al. papers, which both focus on two-monitor systems. If there are many monitors, then the approximate methods cited in lines 295-296 / 70-71 are more appropriate (see also discussion in lines 72-75 of the paper).
>
> **Re distribution shift/complex monitors**: See additional experiments mentioned above. Our method works with any monitor that outputs numbers as scores, since then we can sweep over regions to call additional monitors over. Similarly, likelihood ratios can be calculated for any random variable.
>
> **Re attack types**: We do believe that monitor design needs to adapt to the settings they’re in, but that should happen at the monitor level. Our core contribution is the combination mechanism, which helps monitors complement each other but cannot fix issues that all underlying monitors suffer from.
>
> **Re monitor miscalibration**: In a sense, our llama 70B monitor was already slightly miscalibrated. We think that likelihood-based ratios will generally be helpful for monitor calibration. However, severely miscalibrated monitors should probably be avoided instead of “fixed” post hoc with our monitor combination methods.
>
> **Re computational overhead**: We think the overhead during runtime will be minimal. Instead of calculating the log likelihood ratio using kernels for every new input, we could pre-compute what would happen to each region of the monitor score space upfront. Take the fitted policy in Figure 4b, for example. The only overhead this monitor combination adds is along the lines of
> ```
> monitor_1_score = call_monitor_one()
> If monitor_1_score > 6.2:
>     Audit
> Else if monitor_1_score < 5:
>    End
> Else:
>    monitor_2_score = call_monitor_two()
>    monitor_1_score_rounded = round(monitor_1_score, digits = 3) #or some other desired accuracy
>    monitor_2_score_rounded = round(monitor_2_score, digits = 3)
>    If lookup_decision_in_hash_table(monitor_score_1, monitor_score_2) == Audit:
>          Audit
>    Else:
>          End
> ```
> The decision lookup can be completed in constant time. Since our concern is with combining monitors, we think the latency overhead of the monitor calls themselves are out of scope. However, one could imagine measuring the monitor cost in terms of latency, and trading off between different monitors based on their latency.

---

> > ### Comment · Reviewer_Mmt4 · 2025-08-05
> >
> > Thank you for your response, which has addressed most of my concerns, and I have raised my rating accordingly.

---

### Official Review · Reviewer_f693 · 2025-07-03

**Clarity:** 3
**Significance:** 2
**Originality:** 2
**Rating:** 4
**Confidence:** 3

**Summary:**

The paper describes devising a monitor invocation strategy that tries to optimize for cost and recall. The paper shows that using multiple monitors is better than just using one. The paper presents a mathematical framework to formalize the problem as a POMDP and decomposes the policy formulation into two steps: deciding when and which monitors to call, and then determining which outputs to flag. The approach exhaustively searches over a defined class of monitor invocation strategies and uses the Neyman-Pearson lemma. The approach is evaluated on the task of identifying potentially harmful code being generated (viz., code with backdoors) using BigCodeBench.

**Questions:**

How could the approach be adapted for different scenarios? Are there any theoretical or practical limitations to using it for other use cases such as toxicity/bias detection?

How does using AI monitors to provide safety for AI lead to trustworthiness of the overall system?

**Ethical Concerns:**

["NO or VERY MINOR ethics concerns only"]

**Final Justification:**

The paper is addressing an important problem. The author response addressed the main questions in my review.
My review remains mostly positive after reading the other reviews and responses. However, additional experiments with different tasks and more monitors would help sell the practical utility of the approach.

**Limitations:**

yes

**Quality:**

2

**Strengths And Weaknesses:**

+ The paper is addressing an interesting and important problem. It is focusing on the problem of monitoring the outputs of an AI system to determine whether the output is aligned or not in order to decide whether to perform a safety intervention.

+ The paper is generally well written. There is sufficient background provided regarding the problem setting and the solution. The experimental evaluation is clearly defined.

- The approach is fairly straightforward using multiple monitors and using exhaustive search through possible monitor invocations utilizing the Neyman-Pearson lemma. It is not clear whether the technique will be able to detect more sophisticated backdoors than those generated by o1-mini.

- The evaluation is limited. The specific task that is being used is to determine whether the code contains backdoors using BigCodeBench. An evaluation on more extensive tasks would make the paper stronger.

---

> ### Author Rebuttal · Authors · 2025-07-30
>
> We thank reviewers for their comments. We’ll first give a general overview of the paper’s strengths and answer some common objections. Then dive into answering specific questions.
>
> ## Significance of our work
>
> Our paper is focused on optimally combining information from runtime monitors with different cost and performance profiles. We study how to rigorously construct these protocols agnostic to the specific setting. This is an increasingly important problem in AI safety. Since the Neurips submission deadline, OpenAI has announced that they use a combination of two monitors for ChatGPT Agent’s bio-misuse mitigation, and Anthropic researchers put out a new paper on using multiple probe-based monitors with varied costs. However, neither paper presents a principled way of combining monitors:
>
> Cunningham et al.’s “Cost-Effective Constitutional Classifiers via Representation Re-use” tests various probe-based monitors with different overhead computational costs. They combine monitors using the simple heuristic of “use the cheap monitor first, and if suspicious, use the expensive monitor. ” Instead of finding the optimal region to call the second monitor at a set of budgets to sketch out the cost curve, they simply sweep over the threshold on when to call the second monitor to vary spend (Figure 4). Instead of using likelihood ratios to combine the two monitors when two are called, they only use data from the second monitor.
>
> OpenAI’s “ChatGPT Agent System Card” section 5.2.2.2. also mentions the use of a monitoring system consisting of two monitors. Their first monitor checks if the content is related to biology. If so, a second monitor checks whether the response or prompt contains hazardous information. This shows that monitoring protocols constructed from multiple monitors are already used in production AI systems for AI safety.
>
> To be clear, it’s possible that our optimized algorithm will also recommend a final strategy of “use cheap monitor first, then use expensive monitor, and only rely on expensive monitor’s scores if they’re available.” It’s also possible that practitioners have some strong prior over which monitors are more robust and trustworthy, and choose a simpler or regularized strategy in lieu of something fitted to the data like the one we proposed. However, we believe it’s important to showcase how to construct these protocols in a rigorous, data-driven manner.
>
> The existing literature also relies on heuristics to construct these protocols. McKenzie et al.’s “Detecting High-Stakes Interactions with Activation Probes” chooses when to call the second monitor based on how close the first monitor’s score is to the median (as a heuristic measure for uncertainty). Parrack et al.’s “Benchmarking Deception Probes via Black-to-White Performance Boosts” and McKenzie et al. both take a simple weighted average of multiple monitors to make the final auditing decision, as opposed to using likelihood ratio methods. We hope reviewers will reconsider the timeliness and importance of our work given these new papers, and we will incorporate these citations into our paper.
>
> ## New experiment addressing OOD shift and diverse settings
>
> Reviewers also brought up the lack of diverse settings in our experiments, how our “malicious” code from o1-mini is not sufficiently obscured, and how distributional shifts could reduce the performance of the protocol. We do think distributional shift could be a serious issue: if we overfit to training data, we might do worse on real-world deployments. However, the right comparison is “Are we doing better than not fitting at all?” We ran an additional experiment where we took the thresholds fit to the o1-mini generated backdoors on BigCodeBench, and tested the monitoring protocol’s performance on a random sample of backdoored code from Wen et al. “Adaptive Deployment of Untrusted LLMs Reduces Distributed Threats.” These are backdoored code generated using a much more optimized pipeline on the APPS dataset, which acts as a good out-of-distribution test.
>
> While the overall safety scores decrease significantly, we find that the order in which policies do the best remains the same across all experiments. For the main result in Figure six, when the total budget is 1.5, $\Pi_\text{SR}$’s safety decreased from 82% to 48% on the OOD set, but it still outperforms always using both monitors (24%), or using only the 70B model (32% safety) or 8B model (39% safety). The results are also similar in single monitor settings, where $\Pi_\text{NPS}$ outperforms $\Pi_\text{NP}$, which outperforms $\Pi_\text{Naive}$. Thus, for this specific distribution shift, it was better that we had fitted our thresholds to the data. Future work could test distribution shifts that are more likely to occur for specific settings that we’re constructing monitoring protocols for. At runtime, practitioners could also continue to incorporate new data into the training set and tune the thresholds to keep them updated.
>
> Generally, we think the strength of our paper lies in our principled and rigorous theoretical framework. The important comparison is “Does our paper provide a better way of combining monitors compared to heuristic baselines?” Given that most policy classes search over options which include the heuristic baselines (such as “call the cheap monitor first and the expensive monitor if it’s above a certain threshold”), we think the answer is trivially yes.
>
> ## Adding a new guide on how to apply our methods
>
> Finally, reviewer f693 and RTmL have found our work cleanly written (thank you!). Our goal is to provide an easy-to-follow guide to real-world practitioners building monitoring protocols. Given the rise in systems combining probe-based monitors and black box monitors, we added a guide that walks through an example on how to combine probe monitors and black box monitors using data from Parrack et al., making our work even more approachable.
>
> ## Additional responses for reviewer f683
>
> **Re: detect more sophisticated attacks.** We think the primary way to detect more sophisticated attacks should be to improve the quality of the underlying monitors, which we consider to be out of scope for this paper. The main innovation here is how to combine existing monitors given their performances, and we think our method would be the best one current practitioners could do, and would improve safety more than the existing heuristic methods mentioned above.
>
> **Re limited evaluation**: We’ve added an additional out-of-distribution experiment as mentioned above.
>
> **Re different scenarios such as bias/toxicity detection:** We think this is actually a much easier case to apply our methods to compared to AI control. Our method can be used anytime we already have existing monitors for a situation, and it’s stronger when we have readily accessible data that mimics real-world deployment, so we could fit our decision thresholds on our training data knowing that it reflects the real world well. There exist lots of different ways of monitoring for bias or toxicity, and also lots of data.
>
> **Re Trustworthiness of the system**: We think a lot of the gains would depend on the underlying monitors we’re combining. In general, monitoring and applying safety interventions can guard against threats that are easy to define ahead of time, such as hazardous information on bioweapons. Combining multiple monitors can have gains when the monitors differ in cost and when a second monitor can significantly help disambiguate inputs that the first monitor is uncertain about.

---

> > ### Comment · Reviewer_f693 · 2025-08-04
> >
> > Thank you for your detailed response.

---

> > > ### Comment · Area_Chair_qYaZ · 2025-08-06
> > >
> > > Dear reviewer, would you comment about the specific points in the rebuttal and how they affect your view of the submission? Please comment on the effect of that on your ratings.

---

> > > > ### Comment · Reviewer_f693 · 2025-08-06
> > > >
> > > > The response answers the questions I had. My view continues to remain generally positive after the responses, and I will maintain my current score.

---

### Note · Authors · 2025-08-15

In our paper, we study an important and timely problem: How to combine multiple runtime AI monitors with different cost and performance profiles. Monitor protocols with multiple monitors are already used today in production environments, such as ChatGPT Agent. However, existing protocols and academic work do not use principled methods to combine these monitors, relying instead on heuristics. In our paper, we outline a rigorous mathematical framework for constructing monitor protocols and demonstrate empirically that they can lead to safety gains. Reviewers noted that the paper is well-written and clearly explained our core ideas.

We thank the reviewers for their helpful comments. While many reviewers were concerned about the significance of our work, recent news of companies using multiple monitors has affirmed that studying how to construct monitoring protocols is an impactful and important research area. Reviewers were also rightly concerned about issues from distribution shift. If we fit our thresholds to a specific training distribution, these thresholds could be suboptimal in deployment. However, we argue that the right metric is not whether the thresholds are perfectly optimal, but whether they are better than not using empirical data at all. In our additional experiment on an OOD dataset, we showed that our optimized monitor protocol still significantly outperforms naive baselines.

We believe our paper makes a well-scoped contribution to the growing literature on monitoring AIs. We are grateful for the constructive feedback, which has allowed us to strengthen the paper and resolve key initial concerns. Our work provides a practical, data-driven alternative to the ad-hoc methods currently used in industry, and we hope the committee will recognize its value for building safer, more reliable AI systems.

---

### Decision · Program_Chairs · 2025-09-17

**Decision:**

Accept (poster)

**Comment:**

The paper examines the challenges of efficiently integrating $k$ monitors. Using all the available monitors would increase the quality, but it would also require additional memory and time.  Better policies are possible, especially if ones tolerate stochastic policies.

The problem is cast as a Neyman-Pearson classification, as monitors tend to have more false-positive errors. The submission demonstrated the technique using LLM for code generation. That’s both surprising and expected. Surprising because they could have used any other ML models, but it’s expected, as AI safety is particularly relevant when using LLMs.

The discussion was effective, leading to two reviewers increasing their ratings. As a result, all four reviewers rate the submission as borderline accept.

There is consensus on the quality of the technical work and the honesty of the contributions. The authors made it clear that their work supports up to a few monitors and requires a non-insignificant amount of data for calibrating the classifiers. The analysis of the regions is insightful.

Presenting it as a POMDP was not received well by a reviewer. I agree that this is not a general POMDP, but it’s a way to formally express the problem. It might be possible to use a probabilistic graphical model.

Personally, I find the story clear and that limitations can be studied in follow-up work.

Despite the consensus of borderline accept, I recommend acceptance.